# Cardiac Involvement in Patients with MELAS-Related mtDNA 3243A>G Variant

**Aino-Maija Vuorinen** [1,*], **Lauri Lehmonen** [1], **Mari Auranen** [2], **Sini Weckström** [3], **Sari Kivistö** [1,†], **Miia Holmström** [1] **and Tiina Heliö** [3]

1    HUS Diagnostic Center, Radiology, University of Helsinki and Helsinki University Hospital, 00029 HUS Helsinki, Finland
2    Clinical Neurosciences, ERN-RND Center, Neurology, University of Helsinki and Helsinki University Hospital, 00029 HUS Helsinki, Finland
3    Heart and Lung Center, ERN GUARD-Heart Center, University of Helsinki and Helsinki University Hospital, 00029 HUS Helsinki, Finland
*    Correspondence: aino-maija.vuorinen@hus.fi
†    Deceased Author.

**Abstract:** Mitochondrial myopathy, encephalopathy, lactic acidosis, and stroke-like episodes (MELAS) syndrome is a rare disease with variable clinical manifestations. MELAS is most often caused by the human mitochondrial DNA (mtDNA) m.3243A>G variant. We describe cardiac magnetic resonance (CMR) imaging findings and clinical features of 22 subjects with the m.3243A>G mutation and endeavored to discover the role of CMR in MELAS cardiomyopathy diagnostics. The clinical symptoms, ECG findings, and laboratory tests were retrospectively collected from the electronic medical record. Ten subjects (46%) had cardiac symptoms, and eighteen subjects (82%) had some clinical symptoms or signs of MELAS. Seventeen subjects (77%) showed cardiac findings compatible with MELAS. An ECG showed a short PR interval in six subjects (27%). Two patients had a first-degree atrioventricular block. Repolarization changes in the ECG were observed in thirteen subjects (59%), whereas left ventricular hypertrophy voltage criteria were only observed in one subject. Patients with ECG abnormalities had a strong link between proBNP value and cardiac tissue composition (T1 relaxation, $p < 0.02$) and showed decreased CMR-based strain ($p < 0.025$). The CMR findings are heterogeneous in subjects with m.3243A>G. Cardiac MELAS may include left ventricular hypertrophy, which mimics sarcomericcardiomyopathy but maypredispose individuals to severe heart failure episodes triggered by acute critical situations. CMR may be used to clarify ECG findings. This study indicates that the genetic testing of MELAS should be considered in new cases of HCM or sudden heart failure phenotypes of unknown etiology.

**Keywords:** MELAS syndrome; cardiac magnetic resonance (CMR); ECG; T1; T2; strain

## 1. Introduction

Mitochondrial myopathy, encephalopathy, lactic acidosis, and stroke-like episodes (MELAS) syndrome is one of the most common mitochondrial diseases with an estimated prevalence of 1–16/100,000 in the adult population [1,2]. The MELAS phenotype can be caused by several human mitochondrial DNA (mtDNA) variants, but the mtDNA mutation m.3243A>G is the underlying cause of MELAS syndrome in most patients [3,4]. The heteroplasmy of the mutation on the organ level varies and may be the explanatory factor of the wide phenotype variation. The m.3243A>G mutation carriers are a heterogeneous

group as clinical manifestations include MELAS syndrome, maternally inherited diabetes, deafness, myoclonic epilepsy with ragged red fibers, Leigh syndrome, or MELAS/KSS (Kearns–Sayre syndrome) overlap [5,6]. Additionally, asymptomatic m.3243A>G mutation carriers exist [7]. The m.3243A>G mutation results in dysfunctional energy production in mitochondria; thus, sequelae in organs with high energy consumption, such as the brain and the heart, are expected [8]. Left ventricular hypertrophy, cardiac arrhythmias, and conduction disorders have been reported in m.3243A>G mutation carriers, but the phenotype is quite heterogeneous. MELAS-related hypertrophic cardiomyopathy (HCM) may predispose individuals to severe heart failure episodes triggered by acute critical situations and, in this respect, clinically differs from sarcomeric HCM [9,10]. Cardiac magnetic resonance (CMR) imaging is a noninvasive imaging modality that allows accurate myocardial characterization without ionizing radiation or restrictions by acoustic windows [11,12]. Only a few studies of CMR imaging findings of MELAS patients have been previously published [13]. Identifying cardiac manifestations in MELAS patients is crucial to effectively treat any episodes of heart failure triggered by acute situations such as sepsis or rapid arrhythmias. In this study, we aimed to report the clinical symptoms and CMR findings of genetically proven MELAS patients who are m.3243A>G variant carriers.

## 2. Materials and Methods

We recruited a total of 22 individuals with the 3243A>G mutation. Five of the participants were relatives from two families. Altogether, 15 patients had a family history of MELAS. We retrospectively collected data on clinical symptoms, ECG findings, and laboratory test results (highest measured proBNP, BNP, TnI, CK, and creatine) from the patients' electronic medical records. Cardiac symptoms of the patients comprise dyspnea, chest pain, or cardiac arrhythmias. MELAS cardiac involvement was considered to include one or more of the following: increased left ventricular wall thickness (11 mm in the absence of aortic stenosis or uncontrolled hypertension), left ventricular systolic dysfunction (ejection fraction < 45%), an elevated proBNP, BNP, or TnI level in the absence of other explanatory causes, ECG abnormalities (abnormal rhythm, PR < 120 ms, PR > 200 ms, QRS interval > 120 ms), left ventricular hypertrophy by modified Cornell criteria or Sokolow–Lyon criteria, or repolarization abnormalities such as inverted or flattened T-waves in the absence of other explanatory conditions. We split the patients into two groups (ECG+ and ECG−) based on ECG abnormalities to investigate subtle differences between the groups. This study was reviewed and approved by the local Institutional Review Board and Ethics Committee (permissions HUS/8/2022 and HUS/256/2023), and all patients provided informed consent.

All patients underwent CMR between March 2019 and March 2020. CMR imaging was performed at 1.5T with the Avanto[fit] system (Siemens Healthcare, Erlangen, Germany), using a 32-channel body receiver coil. Retrospective electrocardiographic gating and breath holding were utilized to maximize temporal resolution and to minimize breathing-induced artifacts. The CMR protocol consisted of balanced steady-state free precession (bSSFP) cine sequences with 30 temporal phases in two-chamber (2CH), three-chamber (3CH), and four-chamber (4CH) views, followed by short-axis (SA) cines covering the entire left ventricle (LV) and right ventricle (RV). Pre-contrast and post-contrast shortened modified look-locker inversion recovery T1 mapping and T2-prepared bSSFP pre-contrast T2 mapping were acquired in the same basal, mid-ventricular, and apical SA planes. Additionally, post-contrast phase-sensitive inversion recovery late gadolinium enhancement (LGE) was acquired in SA, 2CH, and 4CH directions. Gadoterate meglumine (Dotarem) of 0.2 mmol/kg was used as a contrast agent. Post-contrast T1 mapping was acquired 12 min after contrast-agent administration.

CMR image analysis was performed with Medis Suite software v4.0, including QMass v8.1, QMap v2.2, and QStrain v4.0 (Medis Medical Imaging Systems, Leiden, The Netherlands), by two experienced radiologists with over 10 years of expertise in CMR. Cine images were semi-automatically segmented for the calculation of LV and RV volumes and the ejection fraction (EF). The same segmentation was used in the feature tracking analysis of endocardial and mid-myocardial LV global longitudinal strain (GLS), as well as global circumferential strain (GCS) in the 2CH direction. T1 and T2 mapping analysis was performed by manually segmenting the acquired basal, mid-ventricular, and apical SA maps. Global T1 and T2 values were calculated based on the 17-segment model created by the American Heart Association [6]. Pre-contrast and post-contrast T1 maps were used in combination with patient hematocrit to calculate extracellular volume fraction (ECV). PSIR late-gadolinium-enhancement images were analyzed using the full width at half maximum method with a 50% threshold to compute LGE percentage (% of LV volume).

Statistical analysis was performed with IBM SPSS Statistics v29 (IBM Corp., Armonk, NY, USA). Normality of continuous variables was evaluated using the Kolmogorov–Smirnov test. The results are reported as means $\pm$ standard deviations in the case of normally distributed variables and as medians (interquartile ranges) otherwise. Differences in the ECG+/$-$ groups were tested with an independent samples Mann–Whitney U test. Correlations were calculated using Spearman's rank correlation. $p$-values < 0.05 were considered statistically significant.

## 3. Results

### 3.1. Clinical Symptoms and Laboratory Results

Out of the entire sample of 22 patients, 60% showed clinically significant cardiac symptoms, and similarly, 68% showed notable differences in the ECG (Table 1). Fifty percent of the sample had hearing-aid equipment. Furthermore, 14% of the study sample were clinically symptom-free, 81% had clinical MELAS symptoms, and 5% of the sample died during the study.

**Table 1.** Clinical symptoms of the patients (N = 22).

| Parameter | Male N = 6 (27%) | Female N = 16 (63%) | Total N = 22 (100%) |
|---|---|---|---|
| Age (y) | 51.9 ± 12.9 | 51.4 ± 13.4 | 50.9 ± 13.4 |
| BMI (kg/m$^2$ ± SD) | 23.3 ± 4.1 | 23.4 ± 4.6 | 23.2 ± 3.9 |
| Positive family history | 4 (67%) | 11 (69%) | 15 (68%) |
| Asymptomatic n/% | 0 (0%) | 4 (25%) | 4 (18%) |
| Diabetes n/% | 5 (83%) | 6 (38%) | 11 (50%) |
| Migraine or headache | 3 (50%) | 3 (19%) | 6 (27%) |
| Stroke | 1 (17%) | 0 (0%) | 1 (5%) |
| Neurodegenerative findings compatible with MELAS in brain MRI | 1 (17%) | 0 (0%) | 1 (5%) |
| Epilepsy | 1 (17%) | 0 (0%) | 1 (5%) |
| Muscle weakness | 3 (50%) | 8 (50%) | 11 (50%) |
| Ocular involvement | 2 (33%) | 5 (31%) | 7 (32%) |
| Hearing loss | 5 (83%) | 11 (69%) | 16 (72.7%) |
| Elevated fasting lactate level | 3 (50%) | 2 (13%) | 5 (23%) |
| Renal failure | 1 (17%) | 4 (25%) | 5 (23%) |
| Hypertension | 2 (33%) | 6 (38%) | 8 (36%) |
| Coronary artery disease | 2 (33%) | 4 (25%) | 6 (27%) |
| Heart failure | 0 (0%) | 2 (13%) | 2 (9%) |
| Metabolic treatment | 1 (17%) | 1 (6%) | 2 (9%) |

BMI = body mass index; MRI = magnetic resonance imaging.

Clinical laboratory tests of the study population are presented in Table 2. The laboratory test results did not differ significantly between the ECG+/− groups.

**Table 2.** Clinical laboratory parameters of the study population (N = 22).

| Parameter | Patients (N = 22) | ECG+ (N = 15) | ECG− (N = 7) |
| --- | --- | --- | --- |
| TnI (ng/L) | 7.5 (20) | 8 (22) | 15 (120) |
| proBNP (ng/L) | 63 (170) | 76 (285) | 55 (58) |
| BNP (ng/L) | 32 (50) | 23 (39) | 60 (-) |
| Lactate (ng/L) | 2 (3) | 2 (1) | 2 (11) |
| CK (ng/L) | 131 (180) | 151 (146) | 103 (239) |
| Creatine (U/L) | 89 ± 30 | 95 ± 35 | 76 ± 12 |

TnI = troponin-I, normal values: <25 ng/L; proBNP = N-terminal prohormone of brain natriuretic peptide, normal values: men < 50 years < 84 ng/L, women < 50 years < 155 ng/L, men 50–65 years < 194 ng/L, and women 50–65 years < 222 ng/L; BNP = brain natriuretic peptide, normal values: men 45–54 years < 84 ng/L, women 45–54 years < 169 ng/L, men 55–64 years < 247 ng/L, and women 55–64 years < 161 ng/L; CK = creatine kinase, normal values: men 18–49 years 50–400 U/L, women > 18 years, 35–210 U/L, and men > 50 years 40–280 U/L.

*3.2. CMR-Based Volumetry, Tissue Characterization, and Strain*

CMR-based cardiac volumetry, together with patients' demographics, is presented in Table 3. There were no significant differences in demographic or volumetric data in the ECG+/− groups.

**Table 3.** Demographics and volumetric data of the study population (N = 22).

| Parameter | Patients (N = 22) | ECG+ (N = 15) | ECG− (N = 7) |
| --- | --- | --- | --- |
| Age at CMR (years) | 50 ± 13 | 53 ± 12 | 46 ± 16 |
| Number of females, N (%) | 17 (77) | 10 (67) | 6 (86) |
| BMI (kg/m$^2$) | 23 (3) | 23 (5) | 22 (2) |
| BSA (m$^2$) | 1.67 ± 0.1 | 1.71 ± 0.1 | 1.61 ± 0.1 |
| MELAS mutation in blood (%) | 28 ± 11 | 22 ± 15 | 30 ± 10 |
| HR (beats per minute) | 74 ± 13 | 74 ± 13 | 75 ± 13 |
| Max LV wall thickness (mm) | 10.6 | 10 (4) | 11 (3) |
| LV EDV (mL/m$^2$) | 75 ± 10 | 76 ± 14 | 78 ± 4 |
| LV ESV (mL/m$^2$) | 28 (6) | 31 (10) | 27 (4) |
| LV EF (%) | 60 ± 7 | 59 ± 7 | 64 ± 2 |
| LV CO (L/(min*m$^2$)) | 3.3 ± 0.7 | 3.2 ± 0.7 | 3.7 ± 1.2 |
| RV EDV (mL/m$^2$) | 68 ± 10 | 66 ± 14 | 77 ± 8 |
| RV ESV (mL/m$^2$) | 24 ± 7 | 24 ± 7 | 27 ± 7 |
| RV EF (%) | 64 ± 7 | 65 ± 8 | 65 ± 6 |
| RV CO (L/(min*m$^2$)) | 3.2 ± 0.7 | 3.1 ± 0.7 | 3.7 ± 0.6 |

CMR = cardiac magnetic resonance; BMI = body mass index; BSA = body surface area; MELAS = mitochondrial myopathy, encephalopathy, lactic acidosis, and stroke-like episodes; HR = heart rate; LV = left ventricular; EDV = end-diastolic volume; ESV = end-systolic volume; EF = ejection fraction; CO = cardiac output; RV = right ventricular.

Quantitative pre-contrast T1 and T2 relaxation times, as well as ECV and LGE results, are presented in Table 4. There were no significant differences in the ECG+/− groups.

Strain analysis results of the study population are presented in Table 5. Significant differences were detected in the mid-myocardial LV GLS and LV GCS between ECG+ and ECG− groups (GLS −20% ± 4% vs. −25% ± 4%, $p = 0.025$; GCS −18% ± 4% vs. −23% ± 5%, $p = 0.009$).

**Table 4.** T1, T2, ECV, and LGE results of the study population (N = 22).

| Parameter | Patients (N = 22) | ECG+ (N = 15) | ECG− (N = 7) |
|---|---|---|---|
| Pre-contrast T1 (ms) | 993 ± 66 | 996 ± 79 | 988 ± 24 |
| Pre-contrast T2 (ms) | 47 ± 3 | 46 ± 3 | 48 ± 1 |
| HKR (%) | 42 ± 2 | 43 ± 2 | 41 ± 2 |
| ECV (%) | 28 ± 3 | 28 ± 4 | 27 ± 1 |
| LGE (%) | 0 (7) | 0 (0) | 0 (11) |

HKR = hematocrit; ECV = extracellular volume fraction; LGE = late gadolinium enhancement.

**Table 5.** Strain results of the study population (N = 22).

| Parameter | Patients (N = 22) | ECG+ (N = 15) | ECG− (N = 7) |
|---|---|---|---|
| LV GLS, endocardial (%) | −26 ± 5 | −25 ± 5 | −28 ± 6 |
| LV GLS, mid-myocardial (%) | −22 ± 5 | −20 ± 4 | −25 ± 5 |
| LV GCS, endocardial (%) | −30 ± 6 | −30 ± 7 | −32 ± 6 |
| LV GCS, mid-myocardial (%) | −20 ± 5 | −18 ± 4 | −23 ± 5 |

LV = left ventricular; GLS = global longitudinal strain; GCS = global circumferential strain.

### 3.3. Statistically Significant Correlations

Statistically significant correlations within the entire study population were detected between proBNP and pre-contrast T1 (R = 0.88, $p < 0.001$), pre-contrast T2 (R = 0.70, $p < 0.016$), and mid-myocardial LV GCS (R = 0.56, $p < 0.037$). Additionally, in the ECG+ group, proBNP was significantly correlated with pre-contrast T1 (R = 0.82, $p < 0.02$) and LGE (R = 0.78, $p < 0.021$).

## 4. Discussion

In the present work, we described CMR findings and clinical symptoms in patients with genetically proven MELAS. MELAS is a heterogeneous syndrome, with clinical symptoms ranging from neurological disorders to specific cardiac involvement. We found a strong link between proBNP values and several CMR parameters, as well as decreased global longitudinal strain and global circumferential strain, in MELAS patients with ECG abnormalities (ECG+ group).

We used clinical examinations and both conventional and novel CMR sequences to form a comprehensive examination of our patient sample. Out of the entire sample of 22 patients, 60% showed clinically significant cardiac symptoms, and similarly, 60% showed notable differences in ECG. Fifty percent of the sample had hearing-aid equipment. Notably, 14% of the study sample were clinically symptom-free, 81% had clinical MELAS symptoms, and 5% of the sample died during the study. CMR-based cardiac volumetry, relaxation time mapping, LGE quantification, and strain analysis were all within normal ranges. We found that the proBNP value of the blood had a significant link to cardiac tissue composition (pre-contrast T1 and T2 relaxation times) and global GCS.

We chose to divide our patient sample based on clinical ECG findings to outline the most significant cardiac-based CMR findings within MELAS, as 68% of the study sample had ECG abnormalities. The division revealed minor differences in the mid-myocardial left ventricular global longitudinal strain and circumferential strain. Other CMR-based volumetric parameters and relaxation time mapping values were within normal ranges [14,15], with no significant LGE detected. The proBNP level correlated with T1 and T2 relaxation times in the entire study population, and even more significantly with the findings of the ECG+ subgroup, suggesting that proBNP measurement may be a useful tool for the screening and follow-up of cardiac disease related to MELAS when ECG findings are present. Our study is limited to a small patient sample, N = 22, making the statistical

analysis of the study only referential. Due to the retrospective nature of the study, we had no age- and gender-matched healthy control group to compare our results to.

Previous studies have referred to the phenotype of the m.3243A>G mutation as highly heterogeneous, a phenotypic chameleon, which is why a clear diagnosis is needed for the prognosis of the disease course [16,17]. Our findings indicate that the CMR findings are also heterogeneous in subjects with m.3243A>G. CMR may be used to clarify clinical, echocardiographic, or ECG-based findings when suspecting cardiac involvement of MELAS syndrome. Decreased strain and the strong link between T1 and T2 relaxation and proBNP were our main CMR-based results.

**Author Contributions:** A.-M.V., M.H., and T.H. designed the study; A.-M.V., M.A., S.W., S.K., M.H., and T.H. gathered the data; A.-M.V., L.L., M.A., S.K., M.H., and T.H. analyzed and interpreted the data; A.-M.V., L.L., M.H., and T.H. wrote the manuscript. All authors have read and agreed to the published version of the manuscript.

**Funding:** This research was funded by Aarne Koskelo Foundation, the Finnish Foundation for Cardiovascular Research, the Finnish Medical Foundation, and Special Governmental Subsidy grants (TYH 2022 209 and 2023213).

**Institutional Review Board Statement:** The study was conducted in accordance with the Declaration of Helsinki and approved by the Ethical Review Committee of the Department of Medicine, University of Helsinki (research permissions: HUS/8/2022 (10 March 2022) and HUS/256/2023 (13 December 2023)).

**Informed Consent Statement:** Informed consent was obtained from all subjects involved in the study.

**Data Availability Statement:** The data are available upon reasonable request from the corresponding author and may not become publicly available due to privacy and ethical reasons.

**Acknowledgments:** We thank the patients for their participation in this study. Author MA is a member of the European Reference Network for Rare Neuromuscular Diseases (ERN EURO-NMD).

**Conflicts of Interest:** The authors declare no conflicts of interest. The funders had no role in the design of the study; in the collection, analyses, or interpretation of the data; in the writing of the manuscript; or in the decision to publish the results.

## Abbreviations

The following abbreviations are used in this manuscript:

| | |
|---|---|
| 2CH | two-chamber |
| 3CH | three-chamber |
| 4CH | four-chamber |
| BMI | body mass index |
| BNP | brain natriuretic peptide |
| BSA | body surface area |
| CK | creatine kinase |
| CO | cardiac output |
| CMR | cardiac magnetic resonance |
| ECG | electrocardiogram |
| ECV | extracellular volume fraction |
| EDV | end-diastolic volume |
| EF | ejection fraction |
| ESV | end-systolic volume |
| GCS | global circumferential strain |

| | |
|---|---|
| GLS | global longitudinal strain |
| HKR | hematocrit |
| HR | heartrate |
| LV | left ventricular |
| LGE | late gadolinium enhancement |
| MELAS | mitochondrial myopathy, encephalopathy, lactic acidosis, and stroke-like episodes |
| mtDNA | mitochondrial DNA |
| proBNP | N-terminal prohormone of brain natriuretic peptide |
| RV | right ventricular |
| TnI | troponin-I |

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
