# Peer review of "Cardiac Involvement in Patients with MELAS-Related mtDNA 3243A>G Variant"

_cardiogenetics, doi:10.3390/cardiogenetics15020016_

Round 1
Reviewer 1 Report
Comments and Suggestions for Authors
Vuorinen et al. investigated cardiac involvement in 22 patients carrying the mtDNA m.3243A>G mutation associated with MELAS syndrome, using cardiac magnetic resonance (CMR) imaging. A majority (77%) showed cardiac abnormalities, including decreased myocardial strain and subtle ECG changes, despite largely normal cardiac volumes and function. Elevated proBNP levels correlated significantly with CMR parameters like T1/T2 relaxation times and myocardial strain, especially in patients with ECG abnormalities. The findings suggest that cardiac manifestations in MELAS are heterogeneous and may be underdiagnosed. The authors propose that CMR and genetic testing should be considered in patients with unexplained hypertrophic cardiomyopathy or sudden cardiac dysfunction. Overall, this study is well designed and has some novelty. It provides new insights into the cardiac field.
I only have one question: The author should involve the age- and sex- matched healthy controls to this study. It will help to distinguish the pathological changes from normal conditions.
Author Response
Comment 1: Overall, this study is well designed and has some novelty. It provides new insights into the cardiac field.
Response 1: Thank you for your positive feedback. We appreciate your recognition of the study's contribution to new insights in the cardiac field.
Comment 2: I only have one question: The author should involve the age- and sex- matched healthy controls to this study. It will help to distinguish the pathological changes from normal conditions.
Response 2: Thank you for your valuable suggestion. We acknowledge the importance of an age- and sex-matched healthy control group to distinguish pathological changes from normal conditions. However, due to the retrospective nature of our study, we were unable to include a control group. We have compared our results to previously published normal values, and this is also why we have divided our study population into ECG+ and ECG- groups. We have addressed this limitation in the discussion section and provided an explanation of the potential impact on our findings.
Reviewer 2 Report
Comments and Suggestions for Authors
The nucleotide change A to G at position m.3243 in the mitochondrial tRNA leucine (UUR) gene (MT-TL1) is the most common point mutation reported in association with the MELAS syndrome. The different heteroplasmy in different organs may help explain the wide variation in phenotype. The m.3243A>G mutation is among the well-recognized cause of HCM. Usually cardiac abnormalities reported in m.3243A>G carriers include myocardial abnormalities, arrhythmias, or conduction defects. Myocardial abnormalities include myocardial thickening, hypertrophic cardiomyopathy, dilated cardiomyopathy, myocardial fibrosis, systolic dysfunction, heart failure that should be early diagnosed for a prompt and appropriate treatment. However, only few systematic studies investigating m.3243A>G carriers for cardiac involvement, have been so far published in the literature.
The Authors report the clinical symptoms and CMR findings observed in 22 genetically proven MELAS patients carrying the mt.3243A>G. They subdivided the patients into two groups (ECG+ and 73 ECG-) based on ECG abnormalities, to investigate subtle differences between the groups. As a result, 60% of them showed clinically significant cardiac symptoms, and similarly, 68% showed notable differences in ECG. 81% of the sample had clinical MELAS symptoms, and 5% died during the study. Cardiac involvement consisted of hypertension (36%), coronary artery disease (27%) and heart failure (9%). CMR based cardiac volumes, relaxation 167 time mapping, LGE quantification, and strain analysis were all within normal ranges while the mid-myocardial left ventricular global longitudinal strain and the circumferential strain had only minor differences in the two subgroups (ECG+ and 73 ECG-). Conversely, they found that proBNP blood values had a significant link to cardiac tissue composition (pre-contrast T1, and T2 relaxation times) and global GCS, suggesting that proBNP measurement may be useful tool for screening and follow-up of cardiac disease related to MELAS.
The manuscript is well written and represents the largest sample of individuals with m.3243A > G mutation studied with CMR. Tables are clear.
I have no major critical issues. I would suggest to expand the discussion by commenting and including in the references the following articles:
- Congenit Heart Dis. 2018 Sep;13(5):671-677. doi: 10.1111/chd.12634.
- 2020 Jun;45(4):356-361. doi: 10.1007/s00059-018-4739-6.
- Neurosciences (Riyadh). 2021 Apr;26(2):128-133. doi: 10.17712/nsj.2021.2.20200145.
- Congenit Heart Dis. 2018 Sep;13(5):671-677. doi: 10.1111/chd.12634.
- 2020 Jun;45(4):356-361. doi: 10.1007/s00059-018-4739-6.
- Neurosciences (Riyadh). 2021 Apr;26(2):128-133. doi: 10.17712/nsj.2021.2.20200145.
Author Response
Comment 1: The manuscript is well written and represents the largest sample of individuals with m.3243A > G mutation studied with CMR. Tables are clear.
Response 1: Thank you for your positive feedback. We appreciate your recognition of the manuscript's quality and the clarity of the tables.
Comment 2: I have no major critical issues. I would suggest to expand the discussion by commenting and including in the references the following articles: ...
Response 2: Thank you for your thorough review of our manuscript. We have expanded the discussion and included the suggested literature in the references as per your recommendations.
Round 2
Reviewer 1 Report
Comments and Suggestions for Authors
The authors addressed all my comments. I have no additional questions.